# Simple Detection of DNA Methyltransferase with an Integrated Padlock Probe

**DOI:** 10.3390/bios12080569

**Published:** 2022-07-26

**Authors:** Yuehua Wang, Yingli Han, Fangyu Zhou, Tingting Fan, Feng Liu

**Affiliations:** 1State Key Laboratory of Chemical Oncogenomics, Guangdong Provincial Key Laboratory of Chemical Biology, Tsinghua Shenzhen International Graduate School, Shenzhen 518055, China; wangyueh19@mails.tsinghua.edu.cn (Y.W.); hyl19@mails.tsinghua.edu.cn (Y.H.); zhoufy20@mails.tsinghua.edu.cn (F.Z.); fantt@sustech.edu.cn (T.F.); 2Department of Chemistry, Tsinghua University, Beijing 100084, China; 3National & Local United Engineering Lab for Personalized Anti-Tumor Drugs, Tsinghua Shenzhen International Graduate School, Shenzhen 518055, China

**Keywords:** DNA methyltransferases, padlock probe, rolling circle amplification, fluorescence

## Abstract

DNA methyltransferases (MTases) can be regarded as biomarkers, as demonstrated by many studies on genetic diseases. Many researchers have developed biosensors to detect the activity of DNA MTases, and nucleic acid amplification, which need other probe assistance, is often used to improve the sensitivity of DNA MTases. However, there is no integrated probe that incorporates substrates and template and primer for detecting DNA MTases activity. Herein, we first designed a padlock probe (PP) to detect DNA MTases, which combines target detection with rolling circle amplification (RCA) without purification or other probe assistance. As the substrate of MTase, the PP was methylated and defended against HpaII, lambda exonuclease, and ExoI cleavage, as well as digestion, by adding MTase and the undestroyed PP started RCA. Thus, the fluorescent signal was capable of being rapidly detected after adding SYBR^TM^ Gold to the RCA products. This method has a detection limit of approximately 0.0404 U/mL, and the linear range was 0.5–110 U/mL for M.SssI. Moreover, complex biological environment assays present prospects for possible application in intricacy environments. In addition, the designed detection system can also screen drugs or inhibitors for MTases.

## 1. Introduction

DNA methylation is an important epigenetic process that is strongly involved in gene expressions, chromatin structures, and tumors [1]. Abnormal DNA methylation can cause malignant proliferation [2], tumor metastasis [3], and changes in overall gene expression patterns [4,5]. Studies have shown that abnormal DNA methylation levels can be used as markers for lung and colon cancer. DNA methylation is a process by which the fifth carbon atom of cytosine covalently binds to a methyl group under the catalysis of MTases to form 5-methylcytosine in humans and other mammals. MTases [6] play an important part in the DNA methylation process because of its ability to transfer methyl groups to some bases accurately. Thus, DNA methylation levels are primarily regulated by DNA methyltransferase. It has been shown that DNA MTase is dysregulated in many cancer cells. Therefore, detection of DNA methyltransferase activity is of great significance for the diagnosis and prognosis of related diseases caused by its abnormal activity.

In view of this, many methods have been developed to detect DNA MTase activity, such as colorimetric methods with double-stranded DNA (dsDNA) probes [7], electrochemistry with dsDNA probes and hairpin probes [8,9,10], fluorescence using dsDNA probes [11,12,13], hairpin probes and dumbbell probes [14,15,16], chemiluminescent immunoassays with dsDNA probes [17] and dumbbell probes [17,18,19,20], and surface-enhanced Raman scattering (SERS) using dsDNA probes [21] and hairpin probes [22]. In addition, these probes usually combine with other nucleotide probes to amplify signals, such as hybridization chain reaction (HCR) [23,24], strand displacement amplification(SDA) [25], and rolling circle amplification (RCA) [18], to detect DNA MTase activity precisely. However, other DNA-sequence-assisted nucleic acid amplification methods increase the complexity of experimental protocol design and experimental cost.

In this paper, we designed a PP integrating the function of substrate and amplification to realize the sensitive detection of DNA MTases without the involvement of purification and extra nucleic acid sequences. PP has the following characteristics: (1) it is not only a substrate of MTases, but also a primer for RCA, and does not need any additional auxiliary probes; (2) it does not need to be purified after cyclization. While using MTase, PP with the sequence of 5′-CCGG-3′ was methylated while avoiding endonuclease HpaII, lambda exonuclease, and exonuclease I digestion; thus, the padlock probe remained undamaged to complete RCA. The fluorescence was detected by subsequently adding SYBR^TM^ Gold. In contrast, PP is a target of the endonuclease HpaII, lambda exonuclease, and exonuclease I without MTase, thus producing single nucleotides. Then, we could detect the activity of DNA MTase by observing the fluorescence signal in the system. Our probe can not only act as an enzyme substrate but also spontaneously perform RCA, so it can accurately detect MTase activity. Moreover, the results from detection in complex biological environments and screening of MTase inhibitors show that this protocol is a sensitive and reliable MTase detection strategy. It also provides new possibilities for clinical detection of cancer.

## 2. Materials and Methods

### 2.1. Reagents

In this study, oligonucleotides were synthesized by Sangon Biotechnology Co., Ltd. (Shanghai, China) and HPLC-purified. All synthesized oligonucleotides are listed in Appendix A. Lambda exonuclease, T4 DNA ligase, exonuclease I, Phi29 DNA polymerase, HpaII restriction endonuclease, M.SssI were acquired from New England Biolabs (Beijing, China), and r Taq DNA Polymerase and T4 PNK were obtained from Takara Biotechnology (Dalian, China) Co., Ltd., and 5-Azacytidine (5-Aza) and 5-fluorouracil were purchased from Sigma–Aldrich. Agarose, 40% polyacrylamide (29:1), and nucleic acid dye Gel Red (10,000×) were purchased from Beyotime Biotechnology (Shanghai, China). SYBR^TM^ Gold was obtained from Life Technologies^TM^ (Eugene, OR, USA).

### 2.2. Apparatus

Fluorescence spectra were recorded by using a microplate reader (Tecan Infinite M1000 Pro, Männedorf, Switzerland). Polyacrylamide and agarose gels were imaged by a Molecular Imager Pharos FX^TM^ Plus system (Bio-Rad, Hercules, CA, USA). All the prepared buffers were diluted using ultrapure water, which was acquired from a Millipore water purification system (Milli-Q, Millipore, Burlington, MA, USA).

### 2.3. MTase Activity Detection Procedures

In this work, the detection system included five steps. First, for PP ligation, 30 µL of unconnected PP DNA, including 10 µL of 10 µM PPs, 10 µL of 10 µM PPc, and 10 µL of ultrapure water, was heated to 95 °C for 5 min, followed by annealing for 1 h prior to ligation. Subsequently, unconnected PP was ligated using 2 U of T4 DNA ligase at room temperature overnight in 100 µL 1× T4 DNA ligase buffer. The mixture was heated for 10 min at 65 °C to inactivate T4 DNA ligase. Second, PP was incubated at 37 °C for 1.5 h with various concentrations of M.SssI MTase in 40 µL 1× CutSmart buffer, including 4 µL of 1600 µM S-adenosyl methionine (SAM), 4 µL of CutSmart buffer (10×), and different volumes of ultrapure water. The mixture was then heated for 20 min at 65 °C to inactivate the M.SssI. Third, unmethylated PP was cleaved at 37 °C for 1 h by adding 2 µL of lambda exonuclease and 1 µL of HpaII restriction endonuclease in 50 µL of 1× CutSmart buffer. Fourth, the DNA fragment was digested at 37 °C for 1 h with 0.3 µL of exonuclease I in 60 µL of 1× ExoI buffer and subsequently heated at 80 °C for 20 min. Finally, 6 µL of ExoI digestant mixture was added to 50 µL of 1× rolling circle amplification reaction buffer containing 0.25 µL of Phi29 DNA polymerase (10 U/µL), 4 µL of dNTPs (25 mM), 0.25 µL of BSA (20 mg/mL), 5 µL of RCA buffer (10×), and 34.5 µL of ultrapure water, followed by incubation at 37 °C for 40 min and heating at 65 °C for 20 min.

### 2.4. Activity Detection of M.SssI with SYBR^TM^ Gold

After RCA, the reaction solution was mixed with 1 µL of SYBR^TM^ Gold (100×) and 49 µL of ultrapure water to a final volume of 100 µL. Then, the mixture was subjected to fluorescence measurements.

### 2.5. Gel Electrophoresis

Agarose gel electrophoresis was used to verify the feasibility of the method. RCA products were loaded for electrophoresis on 1% agarose gel, which was prepared by heating a mixture of agarose (0.25 g), 25 mL of 1×TBE solution, and adding nucleic acid dye, and ran at 140 V for 0.5 h in 1× TBE buffer (89 mM Tris-Borate, 2 mM EDTA, pH 8.3) after RCA. Then, the gel was imaged by using a Molecular Imager Pharos FX^TM^ Plus system (Bio-Rad, Hercules, CA, USA). Polyacrylamide gel electrophoresis was performed to verify the cyclization and digestion progress. The methylation [20] and digestion products were loaded onto 12% native-PAGE gels and incubated for 1 h at 150 V in 1× TBE buffer, stained with SYBR^TM^ Gold for 20 min, and then imaged. The composition of 12% native-PAGE included 3 mL of 40% polyacrylamide (29:1), 7 mL of 1× TBE, 10 µL of TEMED, and 100 µL of 10% APS.

### 2.6. Activity Detection of M.SssI in Human Healthy Serum and Selectivity of the Strategy

Various concentrations of M.SssI were added to 10% healthy human serum to detect MTase activity using the detection system described above.

The selectivity assay was implemented using different target BSA, r Taq DNA Polymerase, T4 PNK, and the next procedures for selectivity as described for MTase activity detection.

### 2.7. Evaluation of MTase Inhibitors

To evaluate the effect of inhibitors on other enzymes in the system, 5-Aza or 5-fluorouracil was added to the system after the completion of methylation reaction, so that the reaction system contained inhibitors 5-Aza or 5-fluorouracil in the subsequent enzymatic lysis digestion process and RCA process. In so doing, the total effect of the two inhibitors on other enzymes in the system could be evaluated in the system without adding inhibitors to the control group. Subsequently, several concentrations of 5-Aza and 5-fluorouracil were used during the methylation process to measure the restraint of inhibitors of MTase. Various concentrations of 5-fluorouracil and 5-Aza were incubated with PP in 1× CutSmart buffer at 37 °C for 30 min. Then, 160 µM SAM and 100 U/mL M.SssI were added to the system, followed by incubation for 1.5 h at 37 °C; then, the next experiments were implemented as described above. The relative activity (RA) of M.SssI was obtained according to Equation (1).
RA = (F_i_ − F_0_)/(F_t_ − F_0_)(1)
where F_0_, F_t_, and F_i_ are the fluorescence intensities without DNA MTase, with DNA MTase, and with both DNA MTase and inhibitor, respectively. The IC50 value of the inhibitor was obtained from the curve-fitting equation.

## 3. Results and Discussion

### 3.1. Scheme of M.SssI Activity Detection Using PP and RCA

PP is usually used to detect some nucleic sequences, such as microRNA [26,27], methylated DNA sequences [28,29,30,31], and exosomes [32], but it has not been previously used to detect DNA MTase activity. To sensitively detect MTase, PP with RCA was designed as illustrated in Figure 1.

First, we synthesized a PPc DNA strand with a 3-hydroxyl terminal and a 5-phosphate terminal. On this chain, we designed the PPs complementary part, the methylation site, the HpaII restriction site and 3-hydroxyl terminal, and a 5-hydroxyl terminal. Under the action of T4 DNA ligase, PPc and PPs will form PP by connecting adjacent 5-phosphate and 3-hydroxyl terminal after annealing. PP is methylated using M.SssI and then cleaved specifically with HpaII. The unmethylated PP is cleaved and exposes 5′-phosphoric acid terminal, to be recognized by Lambda exo. Subsequently, cleaved PP was digested by Lambda exo into single chains to reduce background signal, because SYBR^TM^ Gold can bind all nucleic acids. Finally, single chains including PPc and PPs that did not form PP will be digested into mononucleotide by ExoI to reduce background signal. The cleaved nucleotides cannot carry on RCA after adding Phi29 DNA polymerase and generating fluorescence following the addition of SYBR^TM^ Gold. In contrast, PP is protected from HpaII because the HpaII restriction site is methylated by M.SssI, while the complete PP avoids exonuclease digestion. Finally, under the action of Phi29 DNA polymerase, the PP can facilitate RCA and generate fluorescence following the addition of SYBR^TM^ Gold.

### 3.2. Feasibility of M.SssI Verification Using PP and RCA

To demonstrate the feasibility of this strategy, we designed fluorescent and gel electrophoresis experiments. First, as shown in Figure 1A, an extremely weak fluorescence signal was obtained without M.SssI. In contrast, the fluorescence intensity was significantly stronger with M.SssI, which indicated that our probe can accurately identify whether M.SssI exists. Second, native-PAGE was carried out (Figure 1B). In this experiment, Lane a contained PPs DNA, Lane b contained PPc DNA, Lane c contained products of cyclizing PPs and PPc incubating with T4 DNA ligase, Lane d contained PP that had been incubated with endonucleases, exonucleases but not M.SssI, Lane e contained PP that had been incubated with endonucleases, exonucleases, and M.SssI. The result was similar to Lane c because methylated PP can be protected from digestion by endonucleases and exonucleases. The native-PAGE proved that the experimental process was carried out smoothly according to the scheme. To illustrate the products of RCA with or without M.SssI, agarose gel electrophoresis was also implemented (Appendix A). There were more products of RCA with M.SssI than without M.SssI, which yielded almost no products. In conclusion, fluorescence and agarose gel electrophoresis, as well as polyacrylamide gel electrophoresis methods, demonstrated the feasibility of this method.

### 3.3. Optimization of the Experimental Conditions

To shorten the reaction time and obtain the best signal-to-noise ratio (SNR), we optimized the reaction buffer and the enzymatic reaction time. The methylation and cleavage reaction refer to three kinds of enzymes (M.SssI, HpaII, and lambda exo) and three different buffers (1× NEBuffer 2, 1× lambda reaction buffer, 1× CutSmart buffer). The NEBuffer 2 with 1× concentration (1 mM DTT, 10 mM Tris-HCl, 10 mM MgCl_2_, 50 mM NaCl, pH 7.9) was used for the methylation process. 1× CutSmart buffer (20 mM Tris-Ac,10 mM Mg (OAc)_2_, 50 mM KAc, pH 7.9, 100 µg/mL BSA) was used for HpaII digestion. And 1× lambda reaction buffer (2.5 mM MgCl_2_, 50 µg/mL BSA, pH 9.4, 67 mM glycine-KOH) was used for lambda exo digestion. The cleavage of HpaII was the key to the high SNR and sensitivity in the experiment, and reaction buffer could greatly affect the enzyme activity. To obtain the optimal reaction buffer, we researched the assay properties using CutSmart buffer and different buffer mixtures (CutSmart buffer, CutSmart buffer+ NEBuffer 2, NEBuffer 2 + lambda reaction buffer + CutSmart buffer, lambda reaction buffer + CutSmart buffer). As shown in Appendix A, CutSmart buffer obtained the highest F/F_0_ value. Thus, CutSmart buffer was chosen for the following experiments. To obtain the best performance of M.SssI assay, the methylation reaction time was determined. As shown in Appendix A, the fluorescence intensity was enhanced as the reaction time and the fluorescence intensity grew slowly over the reaction time of 1.5 h. In order to shorten the reaction time, 1.5 h was selected. Therefore, methylation reaction time of 1.5 h was used in the following research. The cleavage time of lambda exo and HpaII was optimized. As shown in Appendix A, the fluorescence intensity value decreased with the reaction time, and subsequently reached a plateau at 20 min. Therefore, cleavage time of 20 min is chosen in the following experiments. Finally, the digestion time of ExoI was optimized. As shown in Appendix A, the F/F_0_ reached its highest value at 1 h. Therefore, in the following experiments, the digestion time was determined to be 1 h.

### 3.4. Sensitivity Detection of MTase and Selectivity of the Strategy

According to the conditions confirmed above, various concentrations of M.SssI were added to the system to detect the sensitivity of the designed method. As the M.SssI concentration increased within the range of 0.5 to 110 U/mL, the intensity of fluorescence increased linearly (Figure 2A). The correlation between the fluorescence intensity and MTase concentration is shown in Figure 2B, which exhibited a good linear correlation within the scope of 0.5 to 110 U/mL (R^2^ = 0.996). The derived correlation equation is F = 348.19 [C] + 1426.04 (Figure 2C), in which F is the fluorescence intensity after adding M.SssI, and [C] is the concentration of M.SssI. The limit of detection (LOD) was 0.0404 U/mL, which follows some reported methods (Appendix A) based on the correlation equation and the LOD equation (3 σ/S). The recovery rates of M.SssI, which had concentrations of 0.5 U/mL, 50 U/mL, and 110 U/mL, were calculated in the reaction buffer, and the results are listed in Table 1. The recovery rate ranged from 89.10–105.2%, and the RSD ranged from 8.87–9.34%. The results indicate that this strategy is reliable and comparable to those obtained in previous studies (Appendix A). The obtained results also verified that the strategy is stable.

To verify the selectivity of the scheme, as shown in Figure 2D, 150 U/mL M.SssI, 50 U/mL T4 PNK, 150 U/mL r Taq DNA polymerase, and 10 mg/mL BSA [33] were added to the detection system. Fluorescence signals were observed after adding MTase as PP could be protected by MTase and signals were weak without MTase, which demonstrated the high specificity of the strategy.

### 3.5. Detection of M.SssI Activity in 10% Human Serum

To assess the application of the designed method in a complex biological environment, various concentrations of M.SssI were added to 10% (*v*/*v*) healthy human serum. As shown in Figure 3, as the M.SssI concentration increased, the fluorescence intensity increased slightly (Figure 3A). The linear range of the calibration curve is 0.5 to 110 U/mL (R^2^ = 0.984), which was in agreement with the linear equation obtained from the CutSmart buffer. (Figure 3B), with an LOD of 0.0381 U/mL. The correlation equation was determined to be F = 369.02 [C] + 1248.32, in which F is the fluorescence intensity with M.SssI and [C] is the concentration of M.SssI in 10% human serum. The recovery rates of M.SssI, which had concentrations of 10 U/mL, 50 U/mL, and 70 U/mL, were calculated in the 10% human serum; the results are listed in Table 2. The recovery rate ranged from 96.09–103.50%, and the RSD ranged from 4.06–6.33%. The results indicate that this strategy is reliable and comparable to those obtained in previous studies (Appendix A). The obtained results verified that the strategy can be used stably in a complex biological environment.

### 3.6. Inhibitory Activity Assay of M.SssI Inhibitors

As an important epigenetic mechanism, DNA methylation has a large impact on gene transcription and is related to many diseases [34,35,36,37]. DNA methylation is a dynamic and reversible process. DNA demethylation leads to transcriptional activation and re-expression of silenced genes, providing a new way of thinking about cancer. Therefore, useful DNA MTase inhibitors that can restrain the activity of DNA MTase have received increasing attention. We used the designed strategy to test the inhibitory effect of DNA MTase inhibitors. The MTase inhibitor 5-Aza, which can be directly incorporated into DNA and inhibit DNA methylation, and 5-fluorouracil (5-F) were used in this assay [38]. Different concentrations of 5-Aza and 5-F were added to the system, respectively. Before the inhibition assay, an experiment was carried out to eliminate the probable impact of the detection system (Appendix A), which revealed that inhibitors had lesser impact on the system. After excluding the inhibitory effect of inhibitors on other enzymes in the system, we evaluated the inhibitory effect of inhibitors on M.SssI activity. The results shown in Figure 4B reveal that the activity of M.SssI decreased as the dosages of 5-F and 5-Aza increased (Figure 4B) in a dose-dependent manner. The half-maximal inhibition concentration (IC50) of 5-F was 8.84 µM, which has a good accordance with previous methods (6.0 ± 0.2 µM) [39]. The IC50 was 4.91 µM of 5-Aza, which is consistent with those obtained in previous studies (4.2 µM, 3.33 µM, 3.5 µM) [7,40,41]. These results prove that the designed method can be used to screen DNA MTase inhibitors.

## 4. Conclusions

In summary, we designed a new single integrated padlock probe to detect DNA MTases with amplification rather than dsDNA probe, hairpin DNA probe, or dumbbell probes to detect the activity of DNA MTases. This proposed strategy is sensitive to a range of 0.5 to 110 U/mL and an LOD of 0.0404 U/mL. This method also shows high specificity to DNA MTase. This project has the following advantages: (1) the padlock probe not only is a substrate for DNA MTases but also can initiate RCA without the help of other probes. In other words, the padlock probe is integrated, which combines detection with RCA; (2) this sensor does not involve the purification process because it does not need other purified ring templates; (3) we used endonucleases and exonucleases to cleave the padlock probe to avoid the nonspecific amplification of RCA. In addition, the padlock probe can be utilized in screening MTase inhibitors and measuring them in complex biological samples. Therefore, our method possesses wide potential for screening drugs, early tumor diagnosis, and medical research.

## Data Availability

Not applicable.

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
