# Peer review of "Simple Detection of DNA Methyltransferase with an Integrated Padlock Probe"

_biosensors, 2022, doi:10.3390/bios12080569_

Round 1
Reviewer 1 Report
In this work, Liu and collaborators report on a homogeneous fluorescent bioassay for determining DNA methyltransferase activity as an indicator of cancer, since this enzyme is upregulated in tumor cells. The authors design a padlock probe acting as an enzyme-substrate and the initiator of isothermal rolling circle amplification (RCA). The enzyme activity is directly proportional to the amount of a fluorescent dye intercalated into the long DNA product. Nonspecific RCA is minimized by using endo and exonucleases that digest the unmethylated padlock. This is an interesting work, however, since the methodology does not involve a receptor immobilized onto a solid support, typically the transducer, it cannot be considered a biosensor. In consequence, it could be out of the scope of this journal. What seems certain is that the keyword biosensor is not appropriate.
From an analytical point of view, the significant figures for the slope and the intercept in the calibration curves have to be adjusted conveniently.
Reviewer 2 Report
A new padlock integrated RCA probe for MTases activity determination were designed, and the probing feasibility and sensitivity were studied. It is an original creative work, the results will be interested to other readers, so I suggest publishing this paper on this journal after being minor revised.
1. The clarity of Scheme 1 and Figure 1B is not enough for publishing.
2. Some spaces are not savable, such as in line 212 and 213, page 6.
3. In line 188 page5, “mg/ml” should be “mg/mL”, and “Mg(Ac)2” should be “Mg(OAc)2”.
Round 2
Reviewer 1 Report
According to IUPAC rules, the methodology developed in this work cannot be named a biosensor or just a sensor. The substitution of the word “biosensor” with “sensor” that the authors have done is by no means the solution. The methodology is a bioassay since the receptor is not immobilized.
In the literature, there are examples of serious mistakes due to reviewers who have not done their work as they should (due to ignorance or not dedicating the necessary time). I don't want to contribute to spreading misconceptions so I insist that the authors should remove the words biosensor and sensor from their manuscript before publication in any journal: Biosensors mdpi if the Editor considers that it is within the scope of this journal, or any other reputable journal.
